# Flipping the curriculum for resident didactics: In-Training and certifying examination scores in an internal medicine residency

Luke McCoy[1]*, Ronald Markert[1], Elysha Thoms[2], Olivia Noall[3], Kathryn Burtson[4]

**1** Department of Medicine, Wright State University, Dayton, Ohio, United States of America, **2** Case Western/Metro Health Medical Center, Cleveland, United States of America, **3** Cleveland Clinic Foundation, Cleveland, United States of America, **4** Department of Medicine, Wright State University, Uniformed Services University, Dayton, United States of America

* luke.mccoy@wright.edu

## Abstract

### Purpose

The flipped classroom (FC) model is increasingly used in undergraduate medical education, where it has been associated with improved knowledge acquisition. However, its impact on graduate medical education (GME) remains underexplored. We hypothesized that implementing a flipped didactics curriculum in our internal medicine residency program would improve performance on the Internal Medicine In-Training Examination (IM-ITE) and the American Board of Internal Medicine Certifying Examination (IM-CE).

### Methods

In 2017, we transitioned from a traditional lecture-based (LB) curriculum to a flipped model. Noon conferences were replaced with problem-based learning (PBL) sessions three days per week, with additional active learning formats used on the remaining days. Morning reports were redesigned to deliver foundational content in advance of PBL sessions. The curriculum followed a 13-block annual structure aligned with ABIM content areas. We compared IM-ITE and IM-CE scores of residents who completed the full flipped curriculum to those trained under the prior LB model.

### Results

We analyzed data from 279 residents who completed or were expected to complete training between 2014 and 2024. FC curriculum residents scored significantly higher on all three IM-ITE exams: ITE1 (62.39% vs. 59.53%, p = 0.008), ITE2 (69.78% vs. 66.54%, p = 0.002), and ITE3 (73.42% vs. 71.12%, p = 0.029). IM-CE scores did not significantly differ between groups (FC = 496 vs. LB = 481, p = 0.32). The first-attempt IM-CE pass rate was significantly higher among FC residents (95.5% vs. 84.0%, p = 0.012).

## Conclusions

Residents in the flipped classroom curriculum demonstrated significantly higher IM-ITE scores and first-attempt pass rates on IM-CE compared to those in the lecture-based curriculum. These outcomes were associated with medium effect sizes, suggesting a meaningful educational benefit of the flipped classroom approach in graduate medical education.

---

In the past decade, flipped learning has become increasingly widespread at all levels of education, including medical education. Since its inception in 2000, flipped learning has taken numerous forms in diverse educational environments [1]. Graduate Medical Education (GME) has begun to adopt the flipped classroom (FC), but to date, there is little research into its effectiveness in this setting [2–5].

## Introduction

Residency didactic education has traditionally been lecture based. In contrast to lecturing, what is done in class and as homework are switched in a flipped classroom. Flipped learning is an active and participatory educational methodology. It requires application of knowledge and discussion of learning objectives, representing higher cognitive level processing on Bloom's taxonomy [6]. Flipped classroom, also known as an inverted classroom, minimizes lecture time in favor of learner participation. At FC's core is offloading foundational knowledge acquisition to the pre-classroom space and replacing the classroom portion with active learning experiences. FC is influenced by constructivist and cognitive load learning theories. Learners with differing levels of baseline knowledge may need to expend varying amounts of time and energy to effectively prepare before coming to the classroom. This ensures that all learners come ready with a similar level of baseline knowledge. Furthermore, the pre-classroom component frees the cognitive capacity for peer collaboration to solve complex problems within the classroom. Pre-class learning takes many diverse forms, including lectures, video lectures, podcasts, vodcasts, textbook readings, and journal articles [7,8]. In the classroom portion, formats include team-based learning (TBL), problem-based learning (PBL), multiple choice questions, objective structured clinical examinations (OSCE), and simulation exercises.[2] The FC approach has been associated with increased knowledge acquisition in higher education [9].

An ever-increasing body of literature describes FC adoption in undergraduate medical education. In this setting, flipped learning has been associated with improved knowledge outcomes and learner satisfaction [2,3]. Therefore, integrating FC into GME would not only follow a natural progression in physician training, but could be expected to improve knowledge acquisition and key outcomes of residency training, such as certifying examinations.

Despite this, the adoption of FC in GME has been rather limited. A recent systematic review of major health and social science databases identified only 22 articles.

While resident satisfaction was noted to be generally high, consistent with flipped learning research in other settings, results for learning outcomes were mixed [4]. Furthermore, the number of studies evaluating program-wide flipped learning, and FC's effect on in-training and certification examinations is even lower [10–16].

We hypothesized that widespread implementation of an FC approach in our resident didactics curriculum would result in improved knowledge acquisition and retention. We hoped that this would improve our Program's In Training Examinations (IM-ITE), IM certification examination (IM-CE) scores and the first-attempt IM-CE pass rate.

## Methods

The Wright-Patterson Medical Center Institutional Review Board reviewed this retrospective research and designated it not human subjects research. Therefore, it was not subject to IRB oversight, and consent was not required. Identifying data was temporarily accessed from residency files during the month of December 2021 for data analysis but was not permanently stored. All flipped classroom interventions were planned residency programmatic changes and not part of a research protocol.

Wright State University's Internal Medicine residency program is university-based and is composed evenly of United States Air Force and civilian residents (approximately 75 total residents, 25 each year of training). Both Air Force and civilian residents rotate at a Veteran's Administration hospital, a large community hospital, and an Air Force hospital. In the academic year 2017–2018, the didactic curriculum was overhauled to implement an FC approach. Before this change, for the graduating years 2014–2017, the didactics curriculum was generally lecture-based (LB). Over a year's time, a workgroup, composed of chief residents and core faculty trained in FC and PBL principles, planned the programmatic overhaul. 75 novel PBL sessions were created by the group or by subspecialist faculty under the group's guidance. The basic weekly schedule of didactic content was unchanged. Residents typically meet for noon conferences and morning reports 5 days per week and once weekly for a 2-hour afternoon session. The LB noon conferences were replaced with 75 newly created PBL sessions facilitated by faculty. PBL sessions occur three days per week while the remaining lectures are replaced with other active learning formats (e.g., simulation training, journal club, case-based interactive discussions). Residents work in self-guided small groups comprising a mixture of training levels. They work for 60 minutes answering free-response clinical questions pertinent to the session's topic. Core generalist faculty, chief residents, and subspecialty attendings are facilitators, guiding learning and occasionally debriefing. Each clinical training site uses the same PBL sessions simultaneously, keeping all residents synchronized to the same content schedule. For the critical pre-class component, two interventions were implemented. Residents are tasked to prepare each day by reviewing *Harrison's Principles of Internal Medicine* readings and Medical Knowledge Self-Assessment Program (MKSAP) sections [17,18]. To ensure that residents come to the PBL sessions with the appropriate foundational knowledge, morning report sessions were reimagined to be an opportunity for pre-class content delivery. Three residents collaboratively prepare a 15-minute primer presentation that contains foundational knowledge for the noon conference of the day. They each deliver it simultaneously at their clinical site. All didactics content is organized into 13 four-week blocks. Each block emphasizes 1–2 organ systems following the American Board of Internal Medicine (ABIM) certification examination (IM-CE) blueprint. This is a one-year curriculum that repeats each year of residency. Once per block, an academic half-day session takes place in a TBL format, providing an opportunity for content and assessment focused on topics from the organ system block. Other weekly 2-hour afternoon sessions were generally unchanged from the old curriculum.

Our study examined scores for In-Training Examinations (IM-ITE) and the IM certification examination (IM-CE) for first-time test takers among residents before and after the redesign of the didactics curriculum.

Two hundred and seventy-nine residents who completed or will complete residency between 2014 and 2024 took all or some of the three yearly IM-ITEs and the IM-CE at the time of this writing. Any score obtained by a resident after they had experienced the full 1-year, 13-block FC curriculum was analyzed as an FC score. Fig 1 shows the LB or FC curriculum designation for a given year and exam.

| Completed Residency | ITE1 | ITE2 | ITE3 | IM-CE |
|---|---|---|---|---|
| 2014 | LB | LB | LB | LB |
| 2015 | LB | LB | LB | LB |
| 2016 | LB | LB | LB | LB |
| 2017 | LB | LB | LB | LB |
| 2018 | LB | LB | TR | FC |
| 2019 | LB | TR | FC | FC |
| 2020 | TR | FC | FC | FC |
| 2021 | FC | FC | FC | FC |
| 2022 | FC | FC | FC | |
| 2023 | FC | FC | | |
| 2024 | FC | | | |

**Fig 1. Examinations under the lecture-based curriculum and flipped classroom curriculum.** Green: LB curriculum exclusively. Pink: Transition(TR): FC curriculum less than 3 months before exam (excluded from analyses). Yellow: FC curriculum for more than 12 months before exam.

Residents completing the residency program from 2014 through 2017 experienced the LB curriculum exclusively. Residents completing in 2018 were under the LB curriculum for ITE1 and ITE2 and under the FC curriculum for less than three months before ITE3. Consequently, for 2018 residents, ITE1–2 was the LB curriculum, and IM-CE was the FC curriculum.

ITE3 was not included in the analysis since the exam occurred shortly after the FC curriculum was introduced (less than three months). Due to similar circumstances, ITE2 for 2019 and ITE1 for 2020 residents were also excluded from the analyses.

## Statistical analysis

Means are reported for IM-ITE and IM-CE examination scores and counts and percentages for IM-CE passing rates. The independent samples t-test compared the LB and FC curricula on IM-ITE and IM-CE examination scores. The chi-squared test compared the LB and FC curricula on IM-CE passing rate. Inferences were made at the 0.05 level of significance. Cohen's $d$ was used to establish the effect size of the new curriculum for ITE1–3 and the IM-CE examination. Analyses were conducted using IBM SPSS Statistics 25.0 (IBM, Armonk, NY).

## Results

Table 1 shows that aggregate performance on all exams except for the IM-CE improved after the implementation of the FC curriculum: ITE1 (62.39% vs. 59.53%, p = 0.008), ITE2 (69.78% vs. 66.54%, p = 0.002), and ITE3 (73.42% vs.71.12%, p = 0.029). LB and FC curricula residents did not differ on mean score for IM-CE (FC = 496 vs. LB = 481, p = 0.32).

FC curriculum residents scored higher on all three ITEs: 2.86% on ITE1, 3.24% on ITE2, and 2.30% on ITE3. Table 2 shows that FC residents had a higher first-attempt pass rate than LB residents (85 of 89 [95.5%] vs. 68 of 81 [84.0%], p = 0.012).

## Discussion

We found a modest effect size improvement on in-training examinations and a higher pass rate on the ABIM certifying examination. We did not demonstrate a significant change in IM-CE scores. The relatively small differences in ITE scores were statistically significant due to reasonably large sample sizes and, as often occurs on a knowledge test in a homogeneous group, low variability (i.e., small standard deviations). For these reasons, educational researchers know that effect size is more informative than statistical significance [19]. Effect size is a measure of the magnitude of an experimental

**Table 1. Lecture-based curriculum vs. flipped classroom curriculum for IM-ITE and IM-CE.**

| Curriculum | ITE1[1] | ITE2[1] | ITE3[1] | IM-CE[2] |
|---|---|---|---|---|
| Lecture-based | N = 144; Mean = 59.53 | N = 126; Mean = 66.54 | N = 91; Mean = 71.12 | N = 81; Mean = 481 |
| Flipped classroom | N = 99; Mean = 62.39 | N = 104; Mean = 69.78 | N = 99; Mean = 73.42 | N = 89; Mean = 496 |
| p* | 0.008 | 0.002 | 0.029 | 0.32 |

[1] percent correct

[2] standardized score with national mean = 500

*independent samples t-test

**Table 2. Lecture-based curriculum vs. flipped classroom curriculum: percent passing IM-CE on first attempt.**

| Curriculum | IM-CE |
|---|---|
| Lecture-Based | 68 of 81 (84.0%) |
| Flipped Conference | 85 of 89 (95.5%) |
| p* | 0.012 |

*chi-squared test

treatment (in our study, the introduction of a new residency curriculum). The larger the effect size, the stronger the relationship between the two variables (for our study, the association between FC curriculum and IM-ITE or IM-CE score). In his seminal work, Cohen developed the formula for effect size with the independent samples t-test:

$$\textit{Cohen's d} \; = \; \textit{mean difference for two groups} \; \div \; \textit{pooled standard deviation}$$

The effect sizes for ITE1–3 were 0.35, 0.41, and 0.32, respectively. Based on Cohen, these *d*'s are classified as medium effect sizes. The IM-CE *d* of 0.15 is considered small [20].

Our findings of improved IM-ITE scores and IM-CE pass rate are consistent with flipped learning literature in higher education and in undergraduate medical education [2,8,21–23]. The IM-ITE is important because it is predictive of performance on the IM-CE [24–26],and improved patient care outcomes have been linked with higher IM-ITE scores [27–29]. In addition, the IM-CE is a key component of program evaluation for internal medicine residencies.

We add to the flipped learning literature in graduate medical education in several ways. First, the limited literature that does exist reports on interventions targeting orientations, modules, or only portions of the curriculum. Outcomes generally focus on resident satisfaction with the change or short-term knowledge acquisition [4,30–33] In our residency, nearly all aspects of the didactics curriculum were converted to FC. Second, we present a relatively large data set from 11 consecutive years and report on knowledge retention outcomes that are of high interest to the GME community.

Given their patient care responsibilities, residents have limited time and energy to acquire foundational knowledge outside the classroom. Our residents were instructed to complete preparatory MKSAP sections and textbook chapters. For those who may not have read the assignments, the re-imagined morning report sessions guaranteed delivery of pre-class content. Anecdotally, many residents reported feeling prepared for flipped conference activities and each block's TBL.

Creating our PBL and TBL content required extensive faculty effort from numerous subspecialist and generalist attendings and chief residents. This creative process occurred over a year's time, involving hundreds of hours of question writing, meetings, and individual communication. Such time commitment is a hurdle for converting to an FC curriculum. Also, any faculty unfamiliarity or discomfort with the FC format should be addressed [34]. Maintenance of the curriculum requires ongoing faculty review of course materials. We have managed to maintain the curriculum over the years reported, partially addressing concerns about the sustainability of an FC curriculum.

There were several limitations to our study. First, the educational innovation was developed and implemented in a single internal medicine residency program. Thus, generalizing to other GME settings should be done with caution. Second, resident comments from our online reporting system would have been useful in evaluating our curricular changes. However, their open-ended responses related to the flipped classroom were too limited to be an unbiased measure of resident satisfaction. We also did not examine the effect of demographic characteristics (e.g., sex, race, ethnicity) or standardized testing information before residency. It has been postulated that flipped classrooms and active learning may have extra benefits for underrepresented students, who may perform better in more interpersonal environments with more faculty-learner interaction [35–38]. We did not investigate flipped learning's other claims, including enhanced ability to learn and improvement of interpersonal and communication skills [32,39,40]. The COVID-19 pandemic affected program implementation and may have impacted test scores [41]. All in-person conferences and learning activities were suspended from March 16, 2020 to July 1, 2020, starting with the COVID shutdown in Ohio. The IM-ITE and IM-CE performance may have suffered from this suspension of group sessions, potentially hiding what would have otherwise been an even stronger positive signal from the curricular change. Ultimately, cause and effect between the intervention and outcomes is not absolutely certain due to the large number of variables inherent in this kind of investigation.

Future directions for investigation could include exploration of outcomes related to improved interpersonal and communication skills, impacts on underrepresented learners in the GME space, or the amount of time that residents are capable of investing in pre-class preparation.

## Conclusions

We found a modest effect size improvement on in-training examinations and a higher pass rate on the ABIM certifying examination for flipped classroom curriculum residents compared to earlier lecture-based curriculum residents. Our before and after study of 11 years of an internal medicine residency adds to the growing literature on flipped classroom approaches in GME. Positive knowledge outcomes in a larger study where the entire didactic curriculum was converted to a flipped format reinforce existing literature and provide the basis for further widespread incorporation into GME.

## Author contributions

**Conceptualization:** Luke McCoy, Kathryn Burtson.

**Data curation:** Elysha thoms, Olivia Noall.

**Formal analysis:** Luke McCoy, Ronald markert, Elysha thoms, Olivia Noall, Kathryn Burtson.

**Investigation:** Luke McCoy, Kathryn Burtson.

**Methodology:** Luke McCoy, Ronald markert, Kathryn Burtson.

**Project administration:** Luke McCoy, Kathryn Burtson.

**Supervision:** Luke McCoy, Kathryn Burtson.

**Validation:** Ronald markert.

**Writing – original draft:** Luke McCoy, Ronald markert, Elysha thoms, Olivia Noall, Kathryn Burtson.

**Writing – review & editing:** Luke McCoy, Ronald markert, Elysha thoms, Olivia Noall, Kathryn Burtson.

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
