## [Decision Letter · Decision Letter 0]

PONE-D-23-33225
Flipping the Curriculum for Resident Didactics: In-Training and Certifying Examination Scores in an Internal Medicine Residency
PLOS ONE

Dear Dr. MCCOY,

Thank you for submitting your manuscript to PLOS ONE. After careful consideration, we feel that it has merit but does not fully meet PLOS ONE’s publication criteria as it currently stands. Therefore, we invite you to submit a revised version of the manuscript that addresses the points raised during the review process.

We look forward to receiving your revised manuscript.

Kind regards,

Sally Mohammed Farghaly

Academic Editor

PLOS ONE

2. We note that your Data Availability Statement is currently as follows: [All relevant data are within the manuscript and its supporting information files.]

Reviewers' comments:

Reviewer's Responses to Questions

**Comments to the Author**

1. Is the manuscript technically sound, and do the data support the conclusions?

Reviewer #1: Partly

2. Has the statistical analysis been performed appropriately and rigorously? 

Reviewer #1: Yes

3. Have the authors made all data underlying the findings in their manuscript fully available?

Reviewer #1: Yes

4. Is the manuscript presented in an intelligible fashion and written in standard English?

Reviewer #1: Yes

5. Review Comments to the Author

Reviewer #1: Thank you for submitting this very interesting article entitled Flipping the Curriculum for Resident Didactics: In-Training and Certifying Examination Scores in Internal Medicine Residency. This article hypothesizes that widespread implementation of a flipped curriculum approach in resident didactics would result in improved knowledge acquisition and retention as reflected on standardized examination scores.

Overall the article is good but requires some major revisions. These following suggestions may help the authors:

Abstract:

1. The first sentence remove the word “that”. There are some grammatical/english sentence structure issues. Suggest careful proofreading.

Introduction:

1. please add more information and a better description of “flipped learning”. While traditional classroom education is well-known, most readers would benefit from some additional clarification. Since the period covers the COVID pandemic, some information regarding how the program dealt with this (and how the data was calculated/interpreted) is necessary to address (perhaps in methods and discussion).

2. Third paragraph, please cite where the recent systematic review identifying only 22 articles took place (PubMed?)

Methods:

1. One weakness of the study involves comparing a cohort of three years (2014-2017) to those after the redesign of the curriculum. Are these groups comparable? The figure is very helpful in understanding the distribution, but does not render well in black and white.

2. it seems that in the flipped classroom, residents come much more prepared to discuss the content then in the traditional classroom. This may confound the results found in the study. It would be interesting to know if any pretest was conducted before sessions.

3. there is mention of a 50-minute primer presentation. Is this in the format lecture? In other words, are learners now receiving both a lecture, as well as an interactive time for discussion?

4. Are the educator team (chief residents, attendings, etc..) consistent for each session and are they the same educators from the years before the change in curriculum?

5. The statistical analysis is adequate but detail showing that the groups are comparable/racial/gender differences is suggested

Discussion

1. There is mention of “longer term knowledge retention” but the data does not show this.

2. How many residents actually prepared for the session is unclear and this may affect results. It is hard to believe that all residents, who are so busy, would have time to consistently prepare. Also the effects of self-preparing and reading, the 50-minute primer presentation and the reverse classroom discussion build and reinforce each other. This point may need to be made more specifically.

3. The two groups contain an unequal number of residents, and the demographics of the residents may not be the same. The question as to if the 2 groups are comparable should be further discussed in the limitations.

4. The data shows no difference in im-ce, this should be furthered addressed in the discussion.

5.

6. PLOS authors have the option to publish the peer review history of their article (what does this mean?). If published, this will include your full peer review and any attached files.

Reviewer #1: No

---

## [Author Response · Author response to Decision Letter 1]

21 Jun 2024

Reviewer Comments to Author:

Reviewer #1: Thank you for submitting this very interesting article entitled Flipping the Curriculum for Resident Didactics: In-Training and Certifying Examination Scores in Internal Medicine Residency. This article hypothesizes that widespread implementation of a flipped curriculum approach in resident didactics would result in improved knowledge acquisition and retention as reflected on standardized examination scores.

Overall the article is good but requires some major revisions. These following suggestions may help the authors:

Abstract:

1. The first sentence remove the word “that”. There are some grammatical/english sentence structure issues. Suggest careful proofreading.

“That” was removed from the first sentence and the manuscript peer-reviewed for grammatical issues.

Introduction:

1. please add more information and a better description of “flipped learning”. While traditional classroom education is well-known, most readers would benefit from some additional clarification. Since the period covers the COVID pandemic, some information regarding how the program dealt with this (and how the data was calculated/interpreted) is necessary to address (perhaps in methods and discussion).

More information was added to the introduction on flipped learning.

Citation: Gilboy MB, Heinerichs S, Pazzaglia G. Enhancing student engagement using the flipped classroom. J Nutr Educ Behav. 2015;47(1):109-114. doi:10.1016/j.jneb.2014.08.008

The COVID pandemic has been clarified and reviewed in the discussion:

“Fifth, the COVID-19 pandemic affected program implementation and may have impacted test scores.41 41 All in-person conferences and learning activities were suspended from March 16, 2020 to July 1, 2020, starting with the COVID shutdown in Ohio. Also, IM-ITE and IM-CE performance may have suffered from interrupted organized group sessions, potentially hiding what would have otherwise been an even stronger positive signal from the curricular change.”

2. Third paragraph, please cite where the recent systematic review identifying only 22 articles took place (PubMed?)

King AM, Gottlieb M, Mitzman J, Dulani T, Schulte SJ, Way DP. Flipping the Classroom in Graduate Medical Education: A Systematic Review. J Grad Med Educ. 2019;11(1):18-29. doi:10.4300/JGME-D-18-00350.2

This was a review of multiple databases including pubmed, CINAHL Plus, Embase, Web of Science Core Collection, and ERIC.

This has been clarified in the text.

Methods:

1. One weakness of the study involves comparing a cohort of three years (2014-2017) to those after the redesign of the curriculum. Are these groups comparable? The figure is very helpful in understanding the distribution, but does not render well in black and white.

We ran an independent samples t-test and chi-squared test. Rather than change at the individual level, we assessed for program-level change. We endeavored to pragmatically assess the influence of our flipped classroom implementation.

2. it seems that in the flipped classroom, residents come much more prepared to discuss the content then in the traditional classroom. This may confound the results found in the study. It would be interesting to know if any pretest was conducted before sessions.

That is an interesting point. We collected data for the performance on the validated IM-ITE and ABIM for 3 years pre-, during, and 3 years post- implementation. We did not have a in-house validated pre-assessment. We endeavored to test whether this flipped classroom educational methodology contributed to improved knowledge acquisition. It is probably worth mentioning as well that better pre-class preparation is one of the components of the FC approach.

3. there is mention of a 50-minute primer presentation. Is this in the format lecture? In other words, are learners now receiving both a lecture, as well as an interactive time for discussion?

There is a 15-minute primer presentation that reviews the same learning objectives as the PBL session, in lecture format. This was our method to ensure that the audience had the scaffolding necessary to effectively engage in flipped classroom learning.

The primer is discussed in our methods section:

“Three residents collaboratively prepare a 15-minute primer presentation that contains foundational knowledge for the noon conference of the day.”

We added lectures to our introduction, as well:

“Pre-class learning takes many diverse forms, including lectures, video lectures, podcasts, vodcasts, textbook readings, and journal articles.”

And explored this in our discussion:

“Given their patient care responsibilities, residents have limited time and energy to acquire foundational knowledge outside the classroom. Our residents were instructed to read preparatory articles and textbook chapters, and many reported being ready for flipped conference activities and each block’s TBL. This may represent the impact of re-imagined morning report sessions, which we call the Resident-led Educational Primer. This adaptation was particularly well-received by residents, who voiced that they felt more prepared and able to work collaboratively in the PBL sessions. Furthermore, to our knowledge, this innovation has not been described before in literature. It uniquely and elegantly addresses a principal concern about FC’s feasibility in GME: the difficulty in guaranteeing the crucial pre-class component for busy residents.”

4. Are the educator team (chief residents, attendings, etc..) consistent for each session and are they the same educators from the years before the change in curriculum?

The educator team remained similar. While chiefs change over each academic year, program leadership and core faculty were consistent.

5. The statistical analysis is adequate but detail showing that the groups are comparable/racial/gender differences is suggested

This is an area of future research, that we will explore in our discussion. This data did not align with our research question or hypothesis, when we wrote our research proposal. Our intent way to pragmatically evaluate the impact of a wholesale transition to flipped learning.

“Third, we did not examine the effect of demographic characteristics (e.g., sex, race, ethnicity) or standardized testing information before residency. It has been postulated that flipped classrooms and active learning may have extra benefits for underrepresented students, who may perform better in more interpersonal environments with more faculty-learner interaction”

Discussion

1. There is mention of “longer term knowledge retention” but the data does not show this.

Great point, this was conjecture, we removed the “longer term” qualifier.

2. How many residents actually prepared for the session is unclear and this may affect results. It is hard to believe that all residents, who are so busy, would have time to consistently prepare. Also the effects of self-preparing and reading, the 50-minute primer presentation and the reverse classroom discussion build and reinforce each other. This point may need to be made more specifically.

Thank you for the feedback. We did not measure resident preparation. The 15-minute primer presentation was built into daily schedules, provides necessary scaffolding, and is a unique approach to guaranteeing the pre-classroom component in a GME environment.

“This adaptation was particularly well-received by residents, who voiced that they felt more prepared and able to work collaboratively in the PBL sessions. Furthermore, to our knowledge, this innovation has not been described before in literature. It uniquely and elegantly addresses a principal concern about FC’s feasibility in GME: the difficulty in guaranteeing the crucial pre-class component for busy residents.”

3. The two groups contain an unequal number of residents, and the demographics of the residents may not be the same. The question as to if the 2 groups are comparable should be further discussed in the limitations.

We studied our entire population of residents that trained in our program from 2014-2021. We did not design the study to assess for demographics, but given our somewhat homogenous population, I doubt we would be powered for significance. We did not complete an a-priori power analysis on demographic characteristics, as our research question was population-level.

4. The data shows no difference in im-ce, this should be furthered addressed in the discussion.

Thank you. I have added this to the discussion.

---

## [Decision Letter · Decision Letter 1]

PONE-D-23-33225R1
Flipping the Curriculum for Resident Didactics: In-Training and Certifying Examination Scores in an Internal Medicine Residency
PLOS ONE

Dear Dr. McCoy,

Thank you for submitting your manuscript to PLOS ONE. After careful consideration, we feel that it has merit but does not fully meet PLOS ONE’s publication criteria as it currently stands. Therefore, we invite you to submit a revised version of the manuscript that addresses the points raised during the review process.

We look forward to receiving your revised manuscript.

Kind regards,

Sally Mohammed Farghaly

Academic Editor

PLOS ONE

Journal Requirements:

Reviewers' comments:

Reviewer's Responses to Questions

**Comments to the Author**

1. If the authors have adequately addressed your comments raised in a previous round of review and you feel that this manuscript is now acceptable for publication, you may indicate that here to bypass the “Comments to the Author” section, enter your conflict of interest statement in the “Confidential to Editor” section, and submit your "Accept" recommendation.

Reviewer #1: (No Response)

Reviewer #2: All comments have been addressed

Reviewer #3: (No Response)

Reviewer #4: All comments have been addressed

2. Is the manuscript technically sound, and do the data support the conclusions?

Reviewer #1: Yes

Reviewer #2: Yes

Reviewer #3: Yes

Reviewer #4: Yes

3. Has the statistical analysis been performed appropriately and rigorously? 

Reviewer #1: Yes

Reviewer #2: Yes

Reviewer #3: Yes

Reviewer #4: Yes

4. Have the authors made all data underlying the findings in their manuscript fully available?

Reviewer #1: Yes

Reviewer #2: Yes

Reviewer #3: Yes

Reviewer #4: Yes

5. Is the manuscript presented in an intelligible fashion and written in standard English?

Reviewer #1: Yes

Reviewer #2: Yes

Reviewer #3: Yes

Reviewer #4: Yes

6. Review Comments to the Author

Reviewer #1: Thank you for submitting this revised manuscript. This draft reads much better and most of our suggestions were well integrated into the new manuscript. While this draft is suitable for publication, we have a few minor suggestions.

Introduction:

1. 1st sentence based on lectures” -> Lecture based.

2. “with peers and instructors to apply the same concepts” �  the concept or these concepts

3. “GME has been rather limited”, consider adding “because….(reasons)”

Results:

Consider moving the explanation of effect size and cohen’s d to the discussion.

Discussion:

“professed claims” (consider removing professed).

Consider adding what the future direction of the studies might be to help answer some of the questions that this study raises. It may be interesting to know how long residents spent preparing for the FC at home as well as the consistency of being able to prepare given the time commitments of residency.

Conclusion:

“before and after”.

Figure 1: The black and white table does nor render well. Consider changing “X” for LB, Transition, FC.

Table 1: Can we add range, SEM to the means and N?

Reviewer #2: Thank you for submitting this very valuable paper. This is a very valuable paper that demonstrates the usefulness of Flipping the Curriculum in resident education. I consider that the authors have responded sincerely and adequately to the points raised by the reiewer.

Reviewer #3: Abstract: Succinctly written and reflects the content of the manuscript.

Introduction:

Colloquialisms such as “lay down” in the sentence “Learners must spend time and energy to lay down conceptual frameworks…” should be avoided and replaced with more precise language. Also, with regards to that sentence, I do not understand the authors use of the phrase “conceptual frameworks” in this context. The sentence that follows (“This frees the cognitive capacity classroom learners…”) also needs to be modified as it does not make sense as it is currently written.

The authors have neglected to mention a major advantage of the flipped classroom, which is that, ideally, all learners walk into the classroom with the same baseline level of knowledge. More junior learners will likely need to spend more time with the pre-classroom work than more senior learners, but they should all walk in with the same level of knowledge once that pre-work is completed.

Methods:

The methods of the study are explained clearly.

Do the authors monitor whether the residents are completing the reading and MKSAP questions prior to morning report?

I do wonder what the impetus was for changing to the FC method in the first place. Was it because IM-ITE and IM-CE scores were poor or in need of improvement? If that is the reason for changing to a FC format, then those would be reasonable outcomes to look at. If, however, the change to FC was made for some other reason like low attendance or low satisfaction with the didactics, then those would be better outcomes to measure. It would be helpful if the authors clarified WHY they changed to a FC model, preferably in the introduction, so they can justify their use of exam scores as an outcome.

Results:

The authors should refrain from analyzing the data or making editorial comments (for example, “…educational researchers know that effect size is more informative than statistical significance” or explanation of the definition and importance of effect size) and just report the data in the results section. Commentary should be provided in the discussion section, not here.

Otherwise, the results are presented clearly.

Discussion:

The first paragraph is all over the place and needs to be written more clearly. For example, the authors write “They [residents] must routinely practice on-the-spot critical thinking. Others have shown that…”; who are the “Others” the authors refer to? These sentences don’t seem to follow together naturally.

The authors state “Our residents were instructed to read preparatory articles and textbook chapters, and many reported being ready for flipped conference activities and each block’s TBL.” How do the authors know this? Did they collect data about how prepared the residents felt, or is this anecdotal? If anecdotal, the authors need to make this clear. If there are data to support this statement, the authors need to provide these data.

The authors introduce the novelty of the reimagined morning report sessions, which is great, but seems like it should be an entirely different paper and not addressed here. It detracts from the actual research conducted and presented in this manuscript.

In the final paragraph, I do not know what the authors mean by “…performance may have suffered from interrupted organized group sessions…”.

A major limitation not mentioned by the authors is that you really cannot firmly apply cause and effect to the FC and change in examination scores. There are many, many other variables that could impact examination scores. Were new faculty hired who were excellent teachers? Were the newer residents somehow more inherently motivated to read and learn than the previous residents? Who knows. The FC certainly may have contributed to increased scores, but the cause and effect is certainly not 100%.

Citations: I do not see a #7 in the list of citations, this should be added.

Reviewer #4: The responses to the prior reviewer have been addressed.

There are a few typographical errors to be addressed.

1. Under introduction. The header "Introduction" should be moved before the start of the first paragraph.

2. In the paragraph beginning with "Residency didactic education" in the introduction the 9th sentence "This frees the cognitive capacity" replace the next word "classroom" with "for".

3. In methods 5th sentence, 1st paragraph insert a comma after "workgroup" and one after "faculty".

4. Need space between "typically" and "meet" in the 8th sentence of 1st paragraph

7. PLOS authors have the option to publish the peer review history of their article (what does this mean?). If published, this will include your full peer review and any attached files.

Reviewer #1: No

Reviewer #2: No

Reviewer #3: No

Reviewer #4: No

While revising your submission, please upload your figure files to the Preflight Analysis and Conversion Engine (PACE) digital diagnostic tool, https://pacev2.apexcovantage.com/. PACE helps ensure that figures meet PLOS requirements. To use PACE, you must first register as a user. Registration is free. Then, login and navigate to the UPLOAD tab, where you will find detailed instructions on how to use the tool. If you encounter any issues or have any questions when using PACE, please email PLOS at figures@plos.org. Please note that Supporting Information files do not need this step.<gdiv id="ginger-floatingG-container" style="position: absolute; top: 0px; left: 0px;"><gdiv class="ginger-floatingG ginger-floatingG-closed ginger-floatingG-posdown ginger-floatingG-dirty ginger-floatingG-loading" style="display: block; left: 649.016px; top: 171px; z-index: 51;"><gdiv class="ginger-floatingG-disabled-main"><gdiv class="ginger-floatingG-bar-tool-tooltip ginger-floatingG-bar-tool-tooltip-enable">Enable Ginger</gdiv></gdiv><gdiv class="ginger-floatingG-offline-main"><gdiv class="ginger-floatingG-bar-tool-tooltip">*Cannot connect to Ginger* Check your internet connection

or reload the browser</gdiv></gdiv><gdiv class="ginger-floatingG-enabled-main"><gdiv class="ginger-floatingG-bar"><gdiv class="ginger-floatingG-bar-tool ginger-floatingG-bar-tool-disable"><ga></ga><gdiv class="ginger-floatingG-bar-tool-tooltip">Disable Ginger</gdiv></gdiv><gdiv class="ginger-floatingG-bar-tool ginger-floatingG-bar-tool-rephrase ginger-floatingG-bar-tool-rephrase_big-circle"><ga class="ginger-floatingG-bar-tool-rephrase__btn" id="ginger__floatingG-bar-tool-rephrase__btn">Rephrase</ga><gdiv class="ginger-floatingG-bar-tool-tooltip ginger-floatingG-bar-tool-tooltip_rephrase">Rephrase with Ginger (Cmd+⌥+E)</gdiv></gdiv><gdiv class="ginger-floatingG-bar-tool ginger-floatingG-bar-tool-mistakes"><ga>2</ga><gdiv class="ginger-floatingG-bar-tool-tooltip">Log in to edit with Ginger</gdiv></gdiv></gdiv></gdiv><gdiv class="ginger-floatingG__loading-popup">Ginger is checking your text for mistakes...</gdiv><gdiv class="ginger-floatingG__disabling-popup " style="display: none;"><button class="ginger-floatingG__disabling-popup-button">Disable Ginger in this text field</button><button class="ginger-floatingG__disabling-popup-button">Disable Ginger on this website</button></gdiv><gdiv class="ginger-floatingG-contentPopup" style="display: none;"><gdiv class="ginger-floatingG-contentPopup-wrap-limit">
600/13463 free characters checked.

Go Premium to check longer texts and entire documents
</gdiv></gdiv></gdiv></gdiv>

---

## [Author Response · Author response to Decision Letter 2]

4 Oct 2024

Review Comments to the Author

Reviewer #1: Thank you for submitting this revised manuscript. This draft reads much better and most of our suggestions were well integrated into the new manuscript. While this draft is suitable for publication, we have a few minor suggestions.

Introduction:

1. 1st sentence based on lectures” -> Lecture based.

addressed

2. “with peers and instructors to apply the same concepts” �  the concept or these concepts

Thank you. These sentences have been reworked to address the points raised by reviewer 3.

3. “GME has been rather limited”, consider adding “because….(reasons)”

Certainly, this is intriguing. I have some thoughts but would not feel comfortable speculating here.

Results:

Consider moving the explanation of effect size and cohen’s d to the discussion.

This section has been moved to the discussion.

Discussion:

“professed claims” (consider removing professed).

This has been removed as suggested

Consider adding what the future direction of the studies might be to help answer some of the questions that this study raises. It may be interesting to know how long residents spent preparing for the FC at home as well as the consistency of being able to prepare given the time commitments of residency.

A paragraph addressing these points has been added to the discussion

Conclusion:

“before and after”.

This has been changed as suggested

Figure 1: The black and white table does nor render well. Consider changing “X” for LB, Transition, FC.

These have been changed as suggested

Table 1: Can we add range, SEM to the means and N?

We temporarily accessed the data and do not presently have access to the data.

Reviewer #2: Thank you for submitting this very valuable paper. This is a very valuable paper that demonstrates the usefulness of Flipping the Curriculum in resident education. I consider that the authors have responded sincerely and adequately to the points raised by the reiewer.

Reviewer #3: Abstract: Succinctly written and reflects the content of the manuscript.

Introduction:

Colloquialisms such as “lay down” in the sentence “Learners must spend time and energy to lay down conceptual frameworks…” should be avoided and replaced with more precise language. Also, with regards to that sentence, I do not understand the authors use of the phrase “conceptual frameworks” in this context. The sentence that follows (“This frees the cognitive capacity classroom learners…”) also needs to be modified as it does not make sense as it is currently written.

The authors have neglected to mention a major advantage of the flipped classroom, which is that, ideally, all learners walk into the classroom with the same baseline level of knowledge. More junior learners will likely need to spend more time with the pre-classroom work than more senior learners, but they should all walk in with the same level of knowledge once that pre-work is completed.

Thank you for these points. These sentences have been reworked to avoid colloquialism, improve clarity, and to address this additional major advantage of the flipped classroom.

Methods:

The methods of the study are explained clearly.

Do the authors monitor whether the residents are completing the reading and MKSAP questions prior to morning report?

This is now addressed in the discussion by informing the reader that reports of resident preparedness were anecdotal.

I do wonder what the impetus was for changing to the FC method in the first place. Was it because IM-ITE and IM-CE scores were poor or in need of improvement? If that is the reason for changing to a FC format, then those would be reasonable outcomes to look at. If, however, the change to FC was made for some other reason like low attendance or low satisfaction with the didactics, then those would be better outcomes to measure. It would be helpful if the authors clarified WHY they changed to a FC model, preferably in the introduction, so they can justify their use of exam scores as an outcome.

Excellent point, the last paragraph of the introduction has been altered to reflect our motivations for looking at knowledge acquisition outcomes.

Results:

The authors should refrain from analyzing the data or making editorial comments (for example, “…educational researchers know that effect size is more informative than statistical significance” or explanation of the definition and importance of effect size) and just report the data in the results section. Commentary should be provided in the discussion section, not here.

This section has been moved to the discussion.

Otherwise, the results are presented clearly.

Discussion:

The first paragraph is all over the place and needs to be written more clearly. For example, the authors write “They [residents] must routinely practice on-the-spot critical thinking. Others have shown that…”; who are the “Others” the authors refer to? These sentences don’t seem to follow together naturally.

The first paragraph has been rewritten following the suggestion above to move the commentary on effect size to the discussion.

The authors state “Our residents were instructed to read preparatory articles and textbook chapters, and many reported being ready for flipped conference activities and each block’s TBL.” How do the authors know this? Did they collect data about how prepared the residents felt, or is this anecdotal? If anecdotal, the authors need to make this clear. If there are data to support this statement, the authors need to provide these data.

This has been changed to reflect that this assertion is drawn from anecdotes

The authors introduce the novelty of the reimagined morning report sessions, which is great, but seems like it should be an entirely different paper and not addressed here. It detracts from the actual research conducted and presented in this manuscript.

Thank you for this input. The discussion of these sessions has been scaled back substantially

In the final paragraph, I do not know what the authors mean by “…performance may have suffered from interrupted organized group sessions…”.

This has been clarified

A major limitation not mentioned by the authors is that you really cannot firmly apply cause and effect to the FC and change in examination scores. There are many, many other variables that could impact examination scores. Were new faculty hired who were excellent teachers? Were the newer residents somehow more inherently motivated to read and learn than the previous residents? Who knows. The FC certainly may have contributed to increased scores, but the cause and effect is certainly not 100%.

This point is well received and has been included in the discussion of limitations.

Citations: I do not see a #7 in the list of citations, this should be added.

#7 has been added to the corresponding article.

Reviewer #4: The responses to the prior reviewer have been addressed.

There are a few typographical errors to be addressed.

1. Under introduction. The header "Introduction" should be moved before the start of the first paragraph.

It has been moved as suggested

2. In the paragraph beginning with "Residency didactic education" in the introduction the 9th sentence "This frees the cognitive capacity" replace the next word "classroom" with "for".

This has been changed as suggested

3. In methods 5th sentence, 1st paragraph insert a comma after "workgroup" and one after "faculty".

This has been changed as suggested

4. Need space between "typically" and "meet" in the 8th sentence of 1st paragraph

This has been changed as suggested

---

## [Decision Letter · Decision Letter 2]

PONE-D-23-33225R2
Flipping the Curriculum for Resident Didactics: In-Training and Certifying Examination Scores in an Internal Medicine Residency
PLOS ONE

Dear Dr. McCoy,

Thank you for submitting your manuscript to PLOS ONE. After careful consideration, we feel that it has merit but does not fully meet PLOS ONE’s publication criteria as it currently stands. Therefore, we invite you to submit a revised version of the manuscript that addresses the points raised during the review process.

We look forward to receiving your revised manuscript.

Kind regards,

Branko Andic

Academic Editor

PLOS ONE

Journal Requirements:

Additional Editor Comments (if provided):

Dear authors,

Thank you for the well-done corrections.

Two reviewers have accepted your manuscript in this form.

However, the third reviewer has a few more suggestions.

Please, to what extent could you do these minimal corrections before the final acceptance and correction of your manuscript.

Thank you in advance.

Kind regards,

Branko Andic

Reviewer's suggestions

Thank you for implementing most of the suggested revisions. The paper reads a lot better and is well organized. Some suggestions:

Abstract: third line impact, conclusion "small-medium" consider revising.

Methods: is this 25 residents per year or for the 3 years?, and at an Air Force hospital.

The discussion is much better and so is the explanation of Cohen's d. I would elaborate on why IM-CE didn't change despite stating that IM-ITE performance is predictive if performance on IM-CE.

Reviewers' comments:

Reviewer's Responses to Questions

**Comments to the Author**

1. If the authors have adequately addressed your comments raised in a previous round of review and you feel that this manuscript is now acceptable for publication, you may indicate that here to bypass the “Comments to the Author” section, enter your conflict of interest statement in the “Confidential to Editor” section, and submit your "Accept" recommendation.

Reviewer #1: All comments have been addressed

Reviewer #3: All comments have been addressed

Reviewer #4: All comments have been addressed

2. Is the manuscript technically sound, and do the data support the conclusions?

Reviewer #1: Yes

Reviewer #3: Yes

Reviewer #4: Yes

3. Has the statistical analysis been performed appropriately and rigorously? 

Reviewer #1: Yes

Reviewer #3: Yes

Reviewer #4: Yes

4. Have the authors made all data underlying the findings in their manuscript fully available?

Reviewer #1: Yes

Reviewer #3: Yes

Reviewer #4: Yes

5. Is the manuscript presented in an intelligible fashion and written in standard English?

Reviewer #1: Yes

Reviewer #3: Yes

Reviewer #4: Yes

6. Review Comments to the Author

Reviewer #1: Thank you for implementing most of the suggested revisions. The paper reads a lot better and is well organized. Some suggestions:

Abstract: third line impact, conclusion "small-medium" consider revising.

Methods: is this 25 residents per year or for the 3 years?, and at an Air Force hospital.

The discussion is much better and so is the explanation of cohen's d. I would elaborate on why IM-CE didn't change despite stating that IM-ITE performance is predictive if performance on IM-CE.

Reviewer #3: The authors did a great job of addressing two rounds of reviewer comments. The manuscript is ready for publication.

Reviewer #4: Thank you for submitting a revised manuscript. All questions and concerns from prior reviews have been addressed. The only minor thing is in the references there are two "17" and no "18" references.

7. PLOS authors have the option to publish the peer review history of their article (what does this mean?). If published, this will include your full peer review and any attached files.

Reviewer #1: No

Reviewer #3: **Yes: **Anne M. Messman, MD, MHPE

Reviewer #4: No

---

## [Author Response · Author response to Decision Letter 3]

18 Feb 2025

Reviewer #1: Thank you for implementing most of the suggested revisions. The paper reads a lot better and is well organized. Some suggestions:

Abstract: third line impact, conclusion "small-medium" consider revising.

Small-medium was changed to modest in an effort to improve clarity. Thank you!

Methods: is this 25 residents per year or for the 3 years?, and at an Air Force hospital.

This has been expanded as (approximately 75 total residents, 25 each year of training)

The discussion is much better and so is the explanation of cohen's d. I would elaborate on why IM-CE didn't change despite stating that IM-ITE performance is predictive if performance on IM-CE.

The phrase, improvements on knowledge assessments has been replaced with more precise language regarding IM-ITE performance and IM-CE pass rate. The IM-CE mean scores did seem to improve as well but did not reach statistical significance. We therefore believe that both standardized knowledge assessments reflect concordant improvements with the flipped classroom curriculum.

---

## [Decision Letter · Decision Letter 3]

PONE-D-23-33225R3
Flipping the Curriculum for Resident Didactics: In-Training and Certifying Examination Scores in an Internal Medicine Residency
PLOS ONE

Dear Dr. McCoy,

Thank you for submitting your manuscript to PLOS ONE. After careful consideration, we feel that it has merit but does not fully meet PLOS ONE’s publication criteria as it currently stands. Therefore, we invite you to submit a revised version of the manuscript that addresses the points raised during the review process.

We look forward to receiving your revised manuscript.

Kind regards,

Jin Su Jeong, Ph.D.

Academic Editor

PLOS ONE

Journal Requirements:

Additional Editor Comments:

Before I can accept this for publication I request one minor revision: please carefully revise the abstract to ensure the findings are clearly presented. There are grammatical errors in places and some ambiguity in how the findings in particular are presented. It may be helpful to ask a trusted colleague to support in revising the abstract to enhance the clarity overall. I make this request as the abstract gives readers the first impression of an article and they are more likely to read and use your work if the abstract flows well and clearly gives all the necessary information.

Reviewers' comments:

Reviewer's Responses to Questions

**Comments to the Author**

1. If the authors have adequately addressed your comments raised in a previous round of review and you feel that this manuscript is now acceptable for publication, you may indicate that here to bypass the “Comments to the Author” section, enter your conflict of interest statement in the “Confidential to Editor” section, and submit your "Accept" recommendation.

Reviewer #1: All comments have been addressed

Reviewer #4: All comments have been addressed

2. Is the manuscript technically sound, and do the data support the conclusions?

Reviewer #1: Yes

Reviewer #4: Yes

3. Has the statistical analysis been performed appropriately and rigorously? 

Reviewer #1: Yes

Reviewer #4: Yes

4. Have the authors made all data underlying the findings in their manuscript fully available?

Reviewer #1: Yes

Reviewer #4: Yes

5. Is the manuscript presented in an intelligible fashion and written in standard English?

Reviewer #1: Yes

Reviewer #4: Yes

6. Review Comments to the Author

Reviewer #1: The authors have done an excellent job at implementing all suggested changes and addressing all reviewer concerns. This shortened version reads well. I believe that this version is ready for publication as is. Some things to consider is to add SEMs on table 1 to show the tightness of the distribution and to remove the colors on page 6 in materials and methods (figure 1 is self explanatory regarding color). Well done.

Reviewer #4: (No Response)

7. PLOS authors have the option to publish the peer review history of their article (what does this mean?). If published, this will include your full peer review and any attached files.

Reviewer #1: No

Reviewer #4: No

---

## [Author Response · Author response to Decision Letter 4]

13 Jun 2025

Additional Editor Comments:

Before I can accept this for publication I request one minor revision: please carefully revise the abstract to ensure the findings are clearly presented. There are grammatical errors in places and some ambiguity in how the findings in particular are presented. It may be helpful to ask a trusted colleague to support in revising the abstract to enhance the clarity overall. I make this request as the abstract gives readers the first impression of an article and they are more likely to read and use your work if the abstract flows well and clearly gives all the necessary information.

The abstract has been rewritten to improve the grammar, clarity, and readability.

Reviewers' comments:

Reviewer's Responses to Questions

Comments to the Author

1. If the authors have adequately addressed your comments raised in a previous round of review and you feel that this manuscript is now acceptable for publication, you may indicate that here to bypass the “Comments to the Author” section, enter your conflict of interest statement in the “Confidential to Editor” section, and submit your "Accept" recommendation.

Reviewer #1: All comments have been addressed

Reviewer #4: All comments have been addressed

2. Is the manuscript technically sound, and do the data support the conclusions?

Reviewer #1: Yes

Reviewer #4: Yes

3. Has the statistical analysis been performed appropriately and rigorously?

Reviewer #1: Yes

Reviewer #4: Yes

4. Have the authors made all data underlying the findings in their manuscript fully available?

Reviewer #1: Yes

Reviewer #4: Yes

5. Is the manuscript presented in an intelligible fashion and written in standard English?

Reviewer #1: Yes

Reviewer #4: Yes

6. Review Comments to the Author

Reviewer #1: The authors have done an excellent job at implementing all suggested changes and addressing all reviewer concerns. This shortened version reads well. I believe that this version is ready for publication as is. Some things to consider is to add SEMs on table 1 to show the tightness of the distribution and to remove the colors on page 6 in materials and methods (figure 1 is self explanatory regarding color). Well done.

Reviewer #4: (No Response)

The colors discussion on page 6 in materials and methods has been removed.

7. PLOS authors have the option to publish the peer review history of their article (what does this mean?). If published, this will include your full peer review and any attached files.

Do you want your identity to be public for this peer review? For information about this choice, including consent withdrawal, please see our Privacy Policy.

Reviewer #1: No

Reviewer #4: No

---

## [Editor Report · Decision Letter 4]

Flipping the Curriculum for Resident Didactics: In-Training and Certifying Examination Scores in an Internal Medicine Residency

PONE-D-23-33225R4

Dear Dr. McCoy,

We’re pleased to inform you that your manuscript has been judged scientifically suitable for publication and will be formally accepted for publication once it meets all outstanding technical requirements.

Kind regards,

Jin Su Jeong, Ph.D.

Academic Editor

PLOS ONE

---

## [Editor Report · Acceptance letter]

PONE-D-23-33225R4

PLOS ONE

Dear Dr. McCoy,

I'm pleased to inform you that your manuscript has been deemed suitable for publication in PLOS ONE. Congratulations! Your manuscript is now being handed over to our production team.

Kind regards,

on behalf of

Dr. Jin Su Jeong

Academic Editor

PLOS ONE